# Nutritional Supplements for the Treatment of Neuropathic Pain

**DOI:** 10.3390/biomedicines9060674

**Published:** 2021-06-13

**Authors:** Khaled M. Abdelrahman, Kevin V. Hackshaw

**Affiliations:** Dell Medical School, University of Texas at Austin, Austin, TX 78712, USA; kabdelrahman@utexas.edu

**Keywords:** neuropathic pain, supplements, zinc, magnesium, vitamin B, vitamin D, curcumin, hypericum, chronic pain

## Abstract

Neuropathic pain affects 7–10% of the population and is often ineffectively and incompletely treated. Although the gold standard for treatment of neuropathic pain includes tricyclic antidepressants (TCAs), serotonin-noradrenaline reuptake inhibitors, and anticonvulsants, patients suffering from neuropathic pain are increasingly turning to nonpharmacologic treatments, including nutritional supplements for analgesia. So-called “nutraceuticals” have garnered significant interest among patients seeking to self-treat their neuropathic pain with readily available supplements. The supplements most often used by patients include vitamins such as vitamin B and vitamin D, trace minerals zinc and magnesium, and herbal remedies such as curcumin and St. John’s Wort. However, evidence surrounding the efficacy and mechanisms of these supplements in neuropathic pain is limited, and the scientific literature consists primarily of preclinical animal models, case studies, and small randomized controlled trials (RCTs). Further exploration into large randomized controlled trials is needed to fully inform patients and physicians on the utility of these supplements in neuropathic pain. In this review, we explore the basis behind using several nutritional supplements commonly used by patients with neuropathic pain seen in rheumatology clinics.

## 1. Introduction

Neuropathic pain is defined as pain caused by a lesion or disease of the somatosensory nervous system and affects 7–10% of the general population [1,2,3]. Neuropathic pain is most often a chronic condition, is associated with anxiety and depression, and negatively impacts quality of life [4,5]. Several pharmacologic therapies have been demonstrated to be effective in neuropathic pain, including tricyclic antidepressants, serotonin-noradrenaline reuptake inhibitors, and the anticonvulsants gabapentin and pregabalin as first-line treatment options in clinical practice [3,6,7]. However, pharmacologic therapies for this condition may not completely relieve neuropathic pain and are associated with significant adverse effects. Additionally, it has been suggested that effective treatment strategies for chronic pain utilize a combination of pharmaceutical and nonpharmaceutical therapies to target pain and its effect on patients’ lives [3,8,9]. Therefore, several nonpharmacologic therapies have been proposed for neuropathic pain, including noninvasive treatments such as exercise therapy, integrated cognitive behavioral therapy, and nutritional supplements. Invasive nonpharmacologic therapies for neuropathic pain include massage therapy, trigger point injections, acupuncture, transcutaneous electrical nerve stimulation (TENS), and motor cortex stimulation (MCS), with varying effects in the attenuation of this chronic pain condition.

So-called “nutraceuticals” and other nonpharmaceutical supplements have gained significant attention in recent years and may serve to work in synergy with existing pharmaceutical-based treatment regimens for combatting chronic neuropathic pain [10,11]. Although the pharmaceutical industry has historically derived its drugs from natural products, nonpharmaceutical natural products and supplements are being increasingly evaluated, with significant advances in high-throughput screening capabilities for nonpharmaceutical natural compounds [12,13]. However, the mechanism of action and efficacy of such nutraceuticals is poorly understood and is the subject of increased attention and investigation to better understand their safety and utility in disease prevention and treatment [14]. Recently, several nutraceuticals have emerged for the treatment of neuropathic pain in a wide range of conditions such as diabetic neuropathy, chemotherapy-related neuropathic pain, and fibromyalgia. The nutritional supplements proposed for the treatment of chronic neuropathic pain include St. John’s Wort (SJW), curcumin, zinc, magnesium, vitamin D, and vitamin B.

Various elegant preclinical animal models, most often in mice, have been developed to study neuropathic pain and leveraged to study the effects of nutritional supplements in neuropathic pain. These models induce neuropathic pain in mice, commonly via peripheral nerve injury models such as axotomy or chronic constriction injury, via drug-induced injury models using anti-cancer agents such as oxaliplatin, or via disease-induced models using streptozotocin to induce diabetic neuropathy [15,16].

However, evidence remains limited on the clinical efficacy of these supplements, and large RCTs are needed to clearly establish the role of each of these compounds in the treatment of neuropathic pain. In this review of the literature, we explore the role of nutritional supplements and other nonpharmaceutical therapies, including zinc, magnesium, vitamin B, vitamin D, curcumin, and SJW, in the treatment of neuropathic pain, including proposed mechanisms of action, efficacy, and evidence supporting their use. The supplements chosen for review were based on the high frequency with which they were inquired about and/or requested by patients seen in university-based clinical practices with a focus on treating patients with neuropathic pain disorders. In Table 1, we summarize the recommended dietary allowance, dietary source, toxic doses, and tested dosage for each of the supplements described in this review. From the initial literature review until April 2021, PubMed, Embase, Google Scholar, and Scopus were searched for applicable studies.

## 2. Zinc for Neuropathic Pain

The mineral zinc, an essential micronutrient, is used for the treatment of a number of diseases such as macular degeneration, sickle cell anemia, alcohol-related liver disease, as well as neuropathic pain [32,33]. The absence of zinc or zinc deficiency is associated with apoptosis, DNA damage, and immune suppression, due in part to the important role of zinc in countering oxidative stress [32,34]. Severe zinc deficiency is implicated in acrodermatitis enteropathica, which can be fatal if untreated [35,36,37]. Furthermore, zinc plays an important physiological role, allowing for the activation of more than 300 genes involved in macromolecule and protein synthesis, as well as cell growth and proliferation [32].

The therapeutic benefit of zinc in the treatment of pain is attributed to its anti-inflammatory properties as a metallothionein [38,39], which is important because inflammation is a driver of chronic pain states including neuropathic pain [40,41]. Zinc has been demonstrated to reduce pain in a number of preclinical models, first in rats, likely due to these anti-inflammatory properties. For example, in a model of rats with induced neuropathic pain, treatment with zinc led to a significant reduction in inflammatory hyperalgesia, with a reduction in the inflammatory biomarker IL-1B as well as nerve growth factor (NGF) [42]. This finding is important because NGF is a major contributor to the peripheral sensory hypersensitivity observed in neuropathic pain [43,44]. Furthermore, NGF has been demonstrated to be an inducer of substance P in experimental models [44,45,46,47]. CSF levels of substance P are increased in fibromyalgia patients, the prototypical neuropathic pain disease model, while serum substance P levels are normal or low in these patients [48]. However, these patients have increased serum levels of angiotensin-converting enzyme (ACE), a zinc-containing dipeptidase responsible for cleaving angiotensin I to angiotensin II, as well as cleavage of substance P into its metabolites [49]. This finding provides further insight into the mechanism by which zinc alters neuropathic pain.

In another study using a similar model, rats with sciatic nerve injury treated with zinc had dose-dependent reductions in thermal hyperalgesia [50]. In mice treated with paclitaxel to model chemotherapy-associated neuropathic pain, pain sensation was reduced through inhibition of the capsaicin receptor TRPV1 [51]. The rat model was also used to demonstrate the important role of zinc in sensory transmission, including its presence in the spinal cord and dorsal root ganglia [52]. More recently, in mouse models, zinc-finger proteins were implicated in the regulation of nociception and pain sensation [53].

However, evidence for the utility of zinc in treating neuropathic pain in human clinical models is limited.

## 3. Magnesium for Neuropathic Pain

Magnesium is a trace metal readily available as an over-the-counter supplement proposed for the treatment of neuropathic pain. It is used in a number of neuropathic pain conditions, including diabetic neuropathy, cancer-related neuropathic pain, chemotherapy-related neuropathy, and postherpetic neuralgia [54,55].

The NMDA receptor, which is blocked by magnesium, has been implicated in the pathogenesis of neuropathic pain and targeted for treatment of the condition [56,57]. The NMDA receptor is involved in central sensitization as well as numerous physiological processes in the central and peripheral nervous systems [58,59]. Thus, the potential use of magnesium for the treatment of neuropathic pain is of interest because magnesium is an antagonist of the NMDA receptor, a key receptor in pain transduction. For example, magnesium has been demonstrated to downregulate spinal cord NMDA receptor phosphorylation in rats with diabetic neuropathic pain and decreased thermal and tactile allodynia in treated rats [60,61]. Additionally, a spinal cord injury neuropathic pain model in rats demonstrated that combined methylprednisolone and magnesium sulfate reduced neuropathic pain measured by thermal hyperalgesia and cold allodynia in rats, suggesting a role for magnesium as a potential adjuvant therapy.

However, clinical data on magnesium in neuropathic pain has yielded mixed results. In a double-blinded RCT of 45 patients suffering from neuropathic pain of various etiologies, oral magnesium supplementation did not significantly improve neuropathic pain compared with the placebo [19]. In another negative trial of 10 patients with peripheral neuropathic pain, magnesium administration did not significantly reduce pain or allodynia, whereas ketamine reduced both pain and allodynia [62]. Similarly, a randomized, double-blind, crossover, placebo-controlled study of 20 patients with neuropathic pain showed no improvement in daily pain intensity in any of the three groups receiving ketamine alone, ketamine plus magnesium, or placebo. Although this study primarily focused on ketamine, it is relevant that adjuvant magnesium did not improve pain [63].

On the contrary, in an RCT of 80 patients with chronic lower back pain with a neuropathic pain component, sequential IV and oral magnesium improved pain intensity compared with the 40 patients in the placebo group [64]. Similarly, a small series of cancer patients with neuropathic pain refractory to opioids had improved pain with intravenous magnesium sulfate [65]. Intravenous magnesium sulfate also reduced neuropathic pain in the short term in patients with neuropathic pain from post-herpetic neuralgia [61,66]

## 4. Vitamin D for Neuropathic Pain

Interest in vitamin D as a therapeutic has increased dramatically in recent years across disciplines [67,68]. Vitamin D is also of significant interest in pain research, as vitamin D deficiency is associated with chronic pain syndromes such as chronic widespread pain, which shares pathophysiological and clinical features with neuropathic pain [69,70,71]. Treatment with vitamin D has been explored extensively in chronic neuropathic pain conditions including fibromyalgia and neuropathic pain in diabetes.

Vitamin D repletion for the treatment of neuropathic pain has been studied in patients with type 2 diabetes mellitus (DM), a condition with high prevalence of vitamin D deficiency [72,73]. Although there is no clear mechanism of action for the treatment of pain with vitamin D, one proposed explanation is that vitamin D is involved in the regulation of inflammatory cytokines [74]. Associations between inflammatory cytokines and vitamin D levels have been previously demonstrated in diabetic neuropathy [75,76]. In one prospective observational study of 51 vitamin-D-deficient DM patients with typical neuropathic pain supplemented with daily vitamin D3 tablets (mean dose of 2059 IU), vitamin D depletion resulted in a significant reduction in neuropathic pain as measured by the McGill pain questionnaire (MPQ) and a visual analog self-report scale (VAS) [77]. In another prospective study of 143 DM patients with painful diabetic neuropathy, patients were treated with a single dose of intramuscular high-dose vitamin D (600,000 IU). This intervention was associated with vitamin D repletion and a significant reduction in pain using the DN4, total pain score, and SFMPQ [78]. These studies were observational and lacked a control group and thus were more subject to bias than randomized controlled trials.

In one randomized controlled trial (RCT) of 184 fibromyalgia patients with diffuse musculoskeletal pain and 104 patients with osteoarthritis (controls), repletion of vitamin D in those with vitamin D levels <20 ng/mL did not reduce pain. Furthermore, vitamin D levels were not associated with pain levels in the study [79]. Conversely, a smaller, randomized, placebo-controlled trial of 57 patients with neuropathic pain secondary to DM found a significant decrease in DN4 pain scores in the treatment group compared with the placebo (*p* = 0.008) [80].

Due to limited evidence, it is difficult to conclude whether vitamin D is an effective treatment for neuropathic pain. Further RCTs testing this treatment, specifically in patients with neuropathic pain, are needed to demonstrate the role of vitamin D in the treatment of neuropathic pain. Such an RCT should also more definitively evaluate the relationship between vitamin D levels and, specifically, neuropathic pain.

## 5. Vitamin B for Neuropathic Pain

B complex vitamins, such as thiamine (B1), pyridoxine (B6), folate (B9), and cyanocobalamin (B12), play a critical role in various physiological processes such as in DNA and RNA synthesis, immunity, and metabolism [81,82,83,84,85]. B vitamins have been hypothesized to alleviate neuropathic pain in diabetic patients, and this hypothesis has been tested in both animal and human models.

For example, a cocktail of vitamins B1, B6, and B12 was found to improve tactile allodynia in diabetic rats, and in the same model, vitamin B6 administration improved sensory nerve conduction in diabetic rats, demonstrating a potential use for B complex vitamins in the treatment of neuropathic pain from diabetes [86]. Furthermore, a systematic review of vitamin B12 or methylcobalamin treatment in painful diabetic neuropathy identified six RCTs assessing pain or somatosensory symptoms with these interventions compared to placebo or baseline [87]. In each of those trials, the intervention significantly improved the somatosensory or neuropathic pain symptoms from baseline when compared with the placebo. Additionally, one study suggested that higher doses of thiamine and pyridoxine (25 and 50 mg/day) result in a more significant reduction in pain from diabetic neuropathy compared with lower doses (1 mg/day of each vitamin) [88].

More recently, Metanx, a combination of the biologically active forms of folate, vitamin B12, and vitamin B6, was tested in a multicenter RCT involving 214 patients with DM and neuropathy. Patients were randomized to Metanx or placebo. Although there was no improvement at 24 weeks’ follow-up in the vibration perception threshold, there was significant improvement in Neuropathy Total Symptom Score (NTSS-6) at 16 weeks, where four of six components of the NTSS-6 are pain. In another study, a daily capsule containing a cocktail of uridine monophosphate, folic acid, and vitamin B12 was administered for two months in 48 patients with peripheral entrapment neuropathy [89]. Patients in the observational study experienced significant reductions in global pain scores and reduced need for analgesic therapy for their pain. These data suggest that a combination of B complex vitamins may be effective in the treatment of neuropathic pain from DM [90,91,92].

Significant interest has surrounded folate as a potential therapeutic for addressing neuropathic pain given its critical role as a methylator in the nervous system. As early as the 1970s, folate was reported as being used in the treatment of neuropathy [93,94]. In one animal model of adult mice with spinal cord injury, treatment with folic acid significantly reduced thermal hyperalgesia compared with control mice [95]. Treatment with folic acid in the same study led to a significant reduction in matrix metalloproteinases (MMP2), which are involved in neuropathic pain induction, compared with controls, suggesting a potential mechanism for the alleviation of pain using folic acid. Metabolically, folate acts as a carrier of one-carbon groups through oxidation pathways, such as the synthesis of purines and pyrimidines. Folate exists throughout the body, although most folate is found in the liver [96,97].

The mechanism of action of vitamin B in treating neuropathic pain and neuropathy is not clear. However, one explanation is that vitamin B12 plays an important role in nerve repair and myelination and may thus improve symptoms of neuropathy, including pain [98,99]. Overall, there are limited randomized trials of vitamin B for neuropathic pain. Larger trials are needed to demonstrate the specific type of vitamin and the magnitude of its effect in treating neuropathic pain.

## 6. Curcumin for Neuropathic Pain

Curcumin is found in the spice turmeric, a member of the ginger family, and is biologically active with anti-inflammatory and antioxidant properties [100,101,102]. It has a long history of being used as a remedy for wounds, aches, and GI disorders, dating as far back as 1900 BC in India [100]. It is hypothesized that the therapeutic mechanism of curcumin is via its inhibition of mitogen-activated protein kinases and suppression of MAP kinase signaling [103,104].

Curcumin for the treatment of neuropathic pain has been studied in various murine models. One study sought to assess the effects of curcumin on neuropathic pain in mice with induced streptozotocin-induced diabetic neuropathy [105]. Four weeks of treatment with curcumin resulted in significant attenuation of thermal hyperalgesia in treated mice. Additionally, treated mice showed decreased release of TNF-a and nitric oxide, suggesting these markers could have played a role in the antinociception observed in the study. In a different mouse model, curcumin treatment resulted in reductions in neuropathic pain, likely attributable to the down-regulation of inflammasome formation and the JAK2-STAT3 cascade in spinal astrocytes, with a concomitant reduction in IL-1B levels [106]. Furthermore, in rats with neuropathic pain induced via chronic constriction injury of the sciatic nerve, early administration of curcumin resulted in a greater reversal of mechanical allodynia at 7 days compared with the placebo, suggesting that curcumin may curb the progression of neuropathic pain to chronic pain following initial nerve insult [107].

Despite extensive studies in preclinical murine models, reporting in the literature of curcumin use in clinical trials for neuropathic pain is limited. In a randomized controlled trial, a multi-ingredient formula containing curcumin phytosome, piperine, and lipoic acid was tested in 141 patients with neuropathic pain, primarily chronic back pain, as an adjunct to the non-steroidal, anti-inflammatory dexibuprofen compared with dexibuprofen alone [108]. The addition of this adjunctive therapy significantly reduced neuropathic pain compared with dexibuprofen alone and did not cause significant adverse effects. Additionally, the use of adjunctive therapy significantly decreased the use of the anti-inflammatory agent by patients in the study. This study further suggests the potential role of nutritional supplements as adjuvant therapy in the treatment of neuropathic pain. However, it is not possible to determine the direct contribution of curcumin to this reduction in pain as the study did not focus on curcumin alone. However, evidence supporting the use of curcumin in neuropathic pain in human subjects is limited and should be the subject of future trials.

## 7. St. Johns’ Wort for the Treatment of Neuropathic Pain

St John’s Wort (SJW), or *Hypericum perforatum*, is a botanical used for its therapeutic properties since antiquity, with its reported benefits in treating disorders of the nervous system dating back to the ancient Greeks and Romans [109,110,111]. SJW has recently been explored as an antinociceptive in neuropathic pain, primarily via multiple rat models [112]. In one model, for example, neuropathic pain in rats was induced either by chronic constriction injury or repeated administration of the chemotherapeutic oxaliplatin. Using the paw pressure test, a measure of the pain threshold, administration of dried SJW extract was demonstrated to reverse mechanical hyperalgesia in the rats. The magnitude of this reversal was similar to that of the TCA amitriptyline, used in the experiment as a reference drug [113]. In a separate model, SJW reversed neuropathic hyperalgesia in mice when used alone and when administered in combination with morphine [114].

Despite interest in its use in the treatment of neuropathic pain, SJW is used primarily in treating depression, where it has been demonstrated to be effective with fewer side effects than standard antidepressants [115,116,117]. While not entirely understood, the proposed hypotheses for the mechanism of SJW in depression include the inhibition of monoamine oxidase as well as the inhibition of amine reuptake at the synapse, similar to the mechanism of action of TCAs as well as other antidepressants [116]. This potentially shared mechanism can help explain why SJW, like TCAs, may be effective in treating neuropathic pain [114,118]. More specifically, it has been shown that SJW components hyperforin and hypericin are responsible for these antinociceptive effects. Another theory for SJW’s mechanism of analgesia revolves around inhibition of protein kinase C pathways, which are implicated in neuropathic pain, by SJW [119].

Despite evidence of its efficacy for treating pain in preclinical models, demonstrations of the clinical efficacy of SJW for neuropathic pain in humans are limited. In one small clinical trial of 54 patients with painful polyneuropathy randomized to SJW or a placebo, there was no difference in pain relief between SJW and the placebo, suggesting that SJW is not an effective treatment for neuropathic pain [31]. Additionally, this study challenges the hypothesis that SJW acts through a similar mechanism to TCAs to relieve pain. However, this was a small study, and larger RCTs are needed to demonstrate whether SJW relieves neuropathic pain. Additionally, hypericum extracted from SJW has been tested in dental neuropathic pain with varying results. Individual case reports suggested the efficacy of hypericum in dental neuropathic pain, whereas a meta-analysis demonstrated high heterogeneity in studies and no significant effect on pain [120]. Similarly, a case report showed that SJW is effective in treating trigeminal neuralgia, an extremely painful, common neuralgia of the face occurring more commonly in women between the ages of 50 and 60 years old [121].

Although generally well-tolerated, certain adverse effects of SJW are documented in the literature, such as drug interactions, hypertensive crisis, and photosensitivity [111,116,122]. SJW is most safe when used as a monotherapy. However, when used in combination with other drugs, important interactions must be considered. SJW is a potent inducer of P-glycoprotein and cytochrome P450 enzymes, and thus lowers the plasma concentration of numerous commonly used drugs from warfarin to tacrolimus [123].

In conclusion, although SJW has been used for a number of neuropathic pain conditions, clinical evidence is anecdotal and highly heterogeneous, emphasizing the need for more rigorous investigations into its use in neuropathic pain treatment.

## 8. Conclusions and Future Directions

Neuropathic pain continues to be a therapeutic challenge, and nonpharmaceutical therapies play an important role in treatment. Nutritional supplements, including trace minerals, vitamins, and herbal products, are increasingly used for the treatment of neuropathic pain. Extensive preclinical animal models have been pivotal in demonstrating potential benefits in neuropathic pain using nutritional supplements and elucidating possible mechanisms of action. Some studies, as described, have explored adjuvant treatment with nutritional supplements in addition to traditional pharmaceuticals. However, in most cases, monotherapy with these supplements is not supported by high-quality evidence in clinical trials. Few clinical trials have been conducted using nutritional supplements for this purpose, and these trials have been small and not powered to robustly investigate the issue at hand.

Overall, it is apparent that there is a significant need for better trials and guidance especially as supplements become increasingly popular. While a number of innovative preclinical models, mostly in mice, has been studied, few studies have effectively tested these nutraceuticals, which are overwhelmingly safe, in human patients. Furthermore, preclinical models are limited for many reasons. For example, neuropathic pain that is induced does not consider the perceptive, emotional, or affective components of pain [124]. Additionally, nutritional treatments effective in mice may not translate to effective treatments in clinical models.

## Figures and Tables

**Table 1 biomedicines-09-00674-t001:** Supplement sources, recommended dosages, and tested doses.

Supplement	Recommended Dietary Allowance		Toxicity Level	Natural Sources	Supplement Doses Available at	Tested Dosages	Ref
	Male	Female					
Zinc	11 mg/d	8 mg/d	100 mg/d	Red meat, poultry, oysters, beans, nuts	30–50 mg	Not tested	[17,18]
Magnesium	400–420 mg/d	310–320 mg/d	5000 mg/d	Leafy vegetables, legumes, nuts, meats	250–500 mg	300 mg orally	[19,20,21,22]
Vitamin D	600–800 IU/d	600–800 IU/d	10,000 IU/d	Egg yolk, fish, liver, UV radiation	400–5000 IU	2000 IU orally	[23,24,25]
Vitamin B9	400 mcg/d dietary folate units (DFE)	400 mcg/dDFE	N/A	Leafy vegetables, nuts, beans, seafood, dairy, grains	400–1000 mcg DFE	Not tested alone	[26,27]
Vitamin B12	2.4 mcg/d	2.4 mcg/d	N/A	Fish, red meat, poultry, eggs, dairy	500–1000 mcg	Not tested alone	[28,29]
Curcumin	N/A	N/A	8000 g/d	Turmeric	500–1000 mg	Not tested alone	[30]
St. John’s Wort	N/A	N/A	Risk of drug interactions	Hypericum Perforatum	100–900 mg	2700 μg	[31]

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
