# Peer review of "Nutritional Supplements for the Treatment of Neuropathic Pain"

_biomedicines, 2021, doi:10.3390/biomedicines9060674_

Round 1

Reviewer 1 Report

Dear Authors,

Thank you for your works. I read it well.

First of all, it was easy to read.

I think it is very suitable for non-professionals to read. I think it would have been nice if the scientific explanation had been given with more clear evidence. I am a scientist who studies neuropathy. I think it would be great to broaden the audience of this journal.

Thank you.

Author Response

We thank the reviewer for the comments regarding the manuscript. In particular, we thank the reviewer for the favorable comments regarding the readability of the manuscript.

Reviewer 2 Report

Line 33 a bracket must be erased;

Line 187 I think the correct word is B1 and not V1.

Author Response

We thank the reviewer for the comments on the manuscript. Both errors pointed out by the reviewer on line 33 and line 187 have been corrected.

Reviewer 2:

Line 33 a bracket must be erased;

Line 187 I think the correct word is B1 and not V1.

Reviewer 3 Report

The use of supplements is very important in neuropathy but unfortunately not well practiced. I feel this article highlight more about this important issue.

Author Response

We thank the reviewer for the kind words regarding the manuscript. In that there were no errors mentioned, there were no revisions made based on the comments.